# Structure and Properties of Organogels Prepared from Rapeseed Oil with Stigmasterol

**DOI:** 10.3390/foods11070939

**Published:** 2022-03-24

**Authors:** Caili Tang, Zheng Wan, Yilu Chen, Yiyun Tang, Wei Fan, Yong Cao, Mingyue Song, Jingping Qin, Hang Xiao, Shiyin Guo, Zhonghai Tang

**Affiliations:** 1College of Food Science and Technology, Hunan Agricultural University, Changsha 410128, China; cailitang@163.com (C.T.); jerrwzy@163.com (Z.W.); eviantyy420@163.com (Y.T.); weifan@hunau.edu.cn (W.F.); qinjingping@hunau.edu.cn (J.Q.); gsy@hunau.edu.cn (S.G.); 2Hunan Engineering Technology Research Center for Rapeseed Oil Nutrition Health and Deep Development, Changsha 410045, China; 3Department of Food Science, University of Massachusetts, Amherst, MA 01003, USA; yiluchen@umass.edu (Y.C.); hangxiao@umass.edu (H.X.); 4College of Food Science, South China Agricultural University, Guangzhou 510642, China; caoyong2181@scau.edu.cn (Y.C.); songmy@scau.edu.cn (M.S.)

**Keywords:** organogel, rapeseed oil, stigmasterol, network structure

## Abstract

This work used the natural ingredient stigmasterol as an oleogelator to explore the effect of concentration on the properties of organogels. Organogels based on rapeseed oil were investigated using various techniques (oil binding capacity, rheology, polarized light microscopy, X-ray diffraction, and Fourier transform infrared spectroscopy) to better understand their physical and microscopic properties. Results showed that stigmasterol was an efficient and thermoreversible oleogelator, capable of structuring rapeseed oil at a stigmasterol concentration as low as 2% with a gelation temperature of 5 °C. The oil binding capacity values of organogels increased to 99.74% as the concentration of stigmasterol was increased to 6%. The rheological properties revealed that organogels prepared with stigmasterol were a pseudoplastic fluid with non-covalent physical crosslinking, and the G’ of the organogels did not change with the frequency of scanning increased, showing the characteristics of strong gel. The microscopic properties and Fourier transform infrared spectroscopy showed that stigmasterol formed rod-like crystals through the self-assembly of intermolecular hydrogen bonds, fixing rapeseed oil in its three-dimensional structure to form organogels. Therefore, stigmasterol can be considered as a good organogelator. It is expected to be widely used in food, medicine, and other biological-related fields.

## 1. Introduction

Traditional hydrogenated fats or saturated fats contain many saturated fatty acids (SFAs) and trans-fatty acids (TFAs) which have an impact on human health [1]. Their excessive intake increases the risk of cardiovascular and cerebrovascular diseases [2], obesity [3], diabetes [4], and other related diseases, making consumers aware of their serious threats to dietary health [5]. Numerous studies have focused on exploring ways to reduce the harmful content of SFAs and TFAs in foods [6,7,8]. Organogels have been considered as an appropriate strategy to reduce SFAs and eliminate TFAs in the diet while increasing the content of unsaturated fats [9].

Organogels are semi-solid systems; their liquid phase is fixed in a thermo-reversible three-dimensional network using various oleogelators which lead to the formation of lipid structures with obvious macroscopic properties (such as oil binding capacity, rheological properties, and thermostability) [10]. As a substitute for saturated fatty acids, and because of their properties, organogels have been widely applied in the food industry and for shaping food products such as cakes, biscuits, meat products, chocolate, and ice cream [11]. Additionally, organogels can be utilized to stabilize and control the release of nutraceuticals and medicines [12]. They can be divided into low-molecular and polymeric organogels by the types of gelators used [13]. The former are called physical organogels formed by the low-molecules which can self-assemble to form supramolecular structures through weak non-covalent bond interactions such as hydrogen bonds, van der Waals forces, hydrophobicity, and π-π interactions [14]. The latter are referred to as chemical organogels, wherein the strong chemical bond between polymer chains form a swelling system with a cross-linked structure in organogels [15]. The physical organogels are more commonly used than the chemical organogels because they can provide a network structure to vegetable oils and are edible. However, as physical organogels have only recently been investigated, detailed information on gelation phenomena and intermolecular interactions is not yet available [16]. Moreover, the types of physical oleogelators are limited, mainly including natural waxes [17], fatty acids, fatty alcohols [18], and compounds of sterols and glutamine [19]. Therefore, a new oleogelator is required for further development.

Stigmasterol (ST), a natural 6-6-6-5 tetracyclic phytosterol [20], is a biosynthesized triterpene sterol. It is commonly found in various plants and deodorized distillates due to the refining of vegetable oils [21]. Recent studies have shown that stigmasterol exhibits a variety of biological activities as an antioxidant [22], anti-inflammatory [23], anti-tumor [24], and anti-diabetic [25]. Stigmasterol (Figure 1) has an amphiphilic structure with a large oleophilic surface and polar OH head group. It can be used as a gelator to immobilize liquid oil in the network structure by the self-assembly method [26].

Rapeseed oil is the second most abundantly produced edible oil in the world and is rich in unsaturated fatty acids such as oleic acid, linoleic acid, and linolenic acid [27]. The type and proportion of fatty acids are more in line with the dietary nutrition standards which can effectively reduce cholesterol and cardiovascular disease [28]. Therefore, we chose rapeseed oil as the base oil to prepare the edible organogels with stigmasterol as the gelator. The effects of different stigmasterol concentrations on the oil binding capacity (OBC), rheological properties, and microstructure of the organogels were researched. The mechanism of gel formation of organogels was studied by polarized light microscopy, X-ray diffraction (XRD), and Fourier transform infrared spectroscopy (FTIR). The results of this study can provide theoretical and technical support for the development and application of phytosterol organogels.

## 2. Materials and Methods

### 2.1. Materials

Stigmasterol (90%) was obtained from Source Leaf Biotechnology (Shanghai, China); commercial grade rapeseed oil (approximately 6% saturated, 58% monounsaturated, and 36% polyunsaturated) was acquired from a local supermarket. The rest of the chemicals and reagents utilized in this experiment were of analytical grade.

### 2.2. Methods

#### 2.2.1. Organogel Preparation

The organogels were prepared by mixing a certain concentration of stigmasterol (1%, 2%, 3%, 4%, 5%, 6%, and 7% (*w*/*w*)) to rapeseed oil. The mixture was heated and stirred at 100 °C for 40 min in oill bath at 200 rpm. After that, the hot mixtures were cooled at 5 °C for 24 h to form a gel. Physical properties of the samples were measured after this storage as described below.

#### 2.2.2. Gelation Temperature Phase Diagram

The organogel samples prepared by mixing stigmasterol at different concentrations (1%, 2%, 3%, 4%, 5%, 6%, and 7% (*w*/*w*)) with rapeseed oil were poured into a serum bottle. After that, they were stored at 5, 10, 15, 20, 25, and 30 °C for 24 h. The self-sustaining ability of the samples was assessed visually by inverting the serum bottle. Samples were categorized as a gel, thickened liquid, or liquid, based on the appearance of behavior [29].

#### 2.2.3. Oil Binding Capacity

By measuring the oil loss of organogels after centrifugation, the oil binding capacities (OBC) of the organogels were obtained. First, the weight of the Eppendorf tube (*a*) was measured and the Eppendorf tube containing 2 g of the melted organogel samples was weighed (*b*); after that, the tubes were stored at 5 °C for 24 h. Second, the tubes were centrifuged at 10,000 rpm for 15 min and then inverted to drain the separated rapeseed oil. The remaining organogel samples in the tube were then weighed (*c*). The oil binding capacity was calculated using the following formula:(1)$$\mathrm{OBC}\left(\%\right)=\frac{\left(c-a\right)}{\left(b-a\right)}\times 100\%$$
where *a* denotes the weight of the empty container, *b* represents the weight of the container containing the primary sample, and *c* denotes the weight of the container containing the sample after centrifugation. All the measurements were conducted in triplicate; the results were reported as mean ± standard deviation (SD).

#### 2.2.4. Rheological Characterization

The rheological properties of the organogels were analyzed by a Kinexus pro advanced rheometer (Malvern Instruments Ltd., Malvern, UK) with a stainless steel cone-plate geometry (40 mm, 1° angle, 1 mm truncation). All the rheological tests were conducted within the linear viscoelastic range. Specifically, the frequency sweep experiments were carried out at 25 °C under a constant strain within the linear viscoelastic domain, ranging from 0.1 Hz to 100 Hz. The temperature sweeps tests were carried out at a constant frequency of 1 Hz and a heating rate of 2 °C/min in the 25–100 °C range. The apparent viscosity was measured with a constant shear strain with varying shear rates (from 0.01 s^−1^ to 100 s^−1^) at 25 °C.

#### 2.2.5. Polarized Light Microscopy

Polarized light microscopy (#CX31., Olympus, Japan) was used to observe organogel crystal morphology. The organogel samples were lightly smeared on a microscope slide and a coverslip was carefully overlaid on the sample. After that, pictures were obtained with 100× magnification using OLYCIA Series Imaging Analysis Software.

#### 2.2.6. X-ray Diffraction Analysis

The XRD pattern was employed to analyze the crystallization patterns forms of the organogels by XRD spectroscopy (SHIMADZU., Kyoto, Japan) with reflection geometry and the Cu Kα radiation (λ = 1.542 Å) operating at 40 kV and 30 mA. The organogel samples were scanned at a scan rate of 2°/min with a 0.02° step size utilizing a 2θ range of 5° to 50°. Each sample was tested in triplicate. The diffractograms were analyzed using MDI Jade 6.0 software (Materials Data Ltd., Livermore, CA, USA).

#### 2.2.7. Fourier Transform Infrared Spectroscopy

The FTIR spectra of the samples were measured using an IRAffinity-1 model FTIR instrument (SHIMADZU, Kyoto, Japan) coupled with an attenuated total reflection (ATR) sampling accessory. The organogel samples, pure stigmasterol, and rapeseed oil were scanned within the 4000–400 cm^−1^ range to explore the interactions of the gel components [30].

#### 2.2.8. Statistical Analysis

All experiments were performed in duplicate or triplicate. The data was analyzed using SPSS 20 (SPSS Inc., Chicago, IL, USA) software, calculating mean value and standard deviation (SD), the results were expressed as mean ±SD. The datasets were subjected to analysis of variance, and Duncan’s multiple range test was used to assess the significant differences between the mean values (a difference of *p* < 0.05 was regarded as substantially different). Furthermore, the figures were drawn using Origin 2018 (OriginLab Corporation, Northampton, MA, USA) for basic data processing and mapping.

## 3. Results and Discussion

### 3.1. Gelation Phase Diagram

Visual observation of appearance was performed to ascertain the gelation of the organogels by simply inverting the serum bottle containing the samples. The systems that did not flow under the influence of gravity were named organogels [31]. Figure 2 shows the appearance and gelling behavior of organogels with different stigmasterol concentrations.

It can be seen from Figure 2 that the formation of organogels was simultaneously affected by gelling temperature and stigmasterol concentration. The samples thickened at all temperatures under the stigmasterol concentration of 1%. When the stigmasterol concentration was ≥2%, the mixtures of stigmasterol and rapeseed oil could form organogels at low temperatures (5 °C). When the stigmasterol concentration was ≥4%, the organogels occurred at room temperature (25 °C). These results showed that the organogels only need a small amount of stigmasterol (2%) to fix rapeseed oil with a 5 °C gelling temperature. With the increase in gelling temperature, the critical gelling concentration of stigmasterol required for the formation of organogels gradually increases. It was possible that the crystallization behavior and crystal structure of stigmasterol in rapeseed oil were extremely sensitive to gelling temperature. The internal structure of organogels by the intermolecular brownian motion decreased with the increasing gelation temperature [26]. According to this result, we could gather the critical concentration of stigmasterol to form organogels at different gelling temperatures. The organogels were prepared within the concentration range of 2–7% at a gelation temperature of 5 °C to further understand the physicochemical and microstructure properties of the organogels.

### 3.2. Physicochemical Properties of Stigmasterol Organogels

The physicochemical properties of stigmasterol organogels were studied by measuring the OBC and rheological properties.

#### 3.2.1. Oil Binding Capacity

The oil binding capacity (OBC) is used to characterize the strength and ability of the organogels to decrease vegetable oil migration [32]. The OBC values of the organogels with different stigmasterol concentrations are shown in Figure 3. The OBC values were increased significantly from 50.74% to 99.74% when the concentration increased from 2% to 6%. It may be that with the increase in stigmasterol concentration, the internal system of the organogels could form more crystal structures through molecular interactions [33]. This further formed a three-dimensional network structure to fix up the rapeseed oil, resulting in significantly increased OBC values. It was worth noting that the OBC value (99.93%) of organogels did not change significantly with a 7% stigmasterol concentration.

The self-assembled structure of stigmasterol may reach its saturation point [34] at 7% concentration with rapeseed oil at 100 °C. These results were similar to those obtained by Zefang Jiang, et al. It has been reported that the formation of organogels highly depends on the ability of the solubility to the gelator, it must be relatively dissolved in solution so that it can crystallize or self-assemble to form a microstructure in a solvent [7].

#### 3.2.2. Rheological Properties

Rheological properties are also important physical and chemical characteristics of organogels. It is essential to understand these rheological properties for the application of organogels. In this experiment, the mechanical stability of the organogel was studied by the oscillatory rheological experiment and the variation law of the apparent viscosity of the organogels with the shear rate was studied by the static rheological experiment.

The viscoelasticity of the sample was reflected by frequency scanning. In rheological analysis, the G’ is the elastic modulus of the sample and the G” is the viscous modulus of the sample. Frequency scanning is utilized to reflect the correlation between the viscoelastic modulus and frequency. If the G’ ˃ G” with the increase in frequency, the sample mainly exhibits elastic deformation, indicating that the sample presents solid state. If the G’ = G”, the sample presents a semi-solid state. If the G’ < G”, the viscosity modulus of the sample mainly has viscous deformation, indicating that the sample presents a liquid state. All organogel samples showed a solid-state behavior with the elastic modulus (G’) higher than the viscosity modulus (G”) within the frequency range of 0.1–100 Hz (see Figure 4). Additionally, the G’ and G” values of organogels were independent of the increase in scanning frequency. These results showed that the organogels prepared from different concentrations of stigmasterol had a good tolerance in the test range of deformation frequency and were formed by a non-covalent physical cross-linked gel network structure [32]. Furthermore, the G’ value was closely related to the stigmasterol concentration, the G’ value increased notably when the concentration was increased from 2% to 6%, but the result was reversed at a concentration of 7%.

Those results in rheology follow the same tendency observed in the oil binding capacity, that is, as the concentration of stigmasterol increased, G’ values and the oil binding capacity increased. However, the G’ value decreased at the stigmasterol concentration of 7%, which could mean that when the stigmasterol concentration reached 7%, the organogel system was under a supersaturated state. This supersaturation may affect the change of crystal structural units in the organogel [35], leading to the decrease in its structural integrity, decreasing the G’ value. It was reported that the supersaturated state could increase the nucleation rate of crystals in the organogels [36], resulting in the formation of more individual networks in the organogels system; however, those crystal structures from different single networks were usually less entangled than the permanent junction of crystal structure in an organogels network. Therefore, the integrity of the structure and the overall mechanical properties decreased with the increasing nucleation rate and the number of structural elements. The results of frequency scanning showed the formation of a gel network and the physical interaction between organgeltor and vegetable oil.

A temperature ramp test of organogels is illustrated in Figure 5 to study the temperature-dependent flow behavior of the stigmasterol organogels. As the scanning temperature increased, the G’ and G” values of organogels were significantly reduced and the critical phase transition temperature (G’ = G”) gradually emerged at the stigmasterol concentration of 2–6%. This result indicates that the organogels showed a viscous behavior at high temperatures in a completely molten state. The critical phase transition temperature increased prominently from 45 °C to 95 °C with the increasing concentration. However, the absence of the critical phase transition temperature was found in the organogels at a 7% stigmasterol concentration, which showed the organogels did not undergo a gel-sol transformation [37]. Therefore, the organogels have high thermal stability with a 7% stigmasterol concentration. This may be because the number of crystals was increased with the increase in the stigmasterol concentration; a higher temperature was needed to destroy the organogel structure [38].

Figure 6 shows that the initial apparent viscosity of organogels increases with the increasing stigmasterol concentration, forming stronger organogel structures. However, the complex viscosity of the organogel samples decreased exponentially as the shear rate was enhanced, reflecting its shear-thinning behavior [39]. This was likely due to the dynamic forces generated in the shearing process causing the fracture of the crystalline structure of stigmasterol organogels [40]. Similar results have been reported in many organogel structures with pseudoplastic properties [40,41,42]. The relationship between the apparent viscosity and shear rate of organogels prepared with different concentrations of stigmasterol is consistent with the power-law equation.
η = Κγ^(n−1)^, 0 < γ ≤ 1(2)
where η denotes the apparent viscosity, γ represents the shear rate, K denotes the flow consistency index, and n is the degree of pseudo-plasticity index [43]. The fitting results (see Table 1) were shown that the organogels with a higher concentration of stigmasterol have higher consistency, the K value reached the maximum at 253.6 Pa·s at a stigmasterol concentration of 7% (see Table 1). More crystals were formed and cross-linked with increasing concentration; therefore, a stronger crystalline structure was provided to the entrapped oil molecules, resulting in higher resistance to shearing. The flow behavior index (n) < 1, between 0.03 to 0.47, indicated that organogel samples were a pseudoplastic fluid in this shear range. The crystal particles’ gradual and orderly arrangement along the direction of shear depolymerization with the shear rate increased [44], which explained why the organogel system became more pseudoplastic and stronger with the increase in stigmasterol concentration. Additionally, the square value of the correlation coefficient (R^2^) of the fitting function was between 0.991 and 0.999, indicating that the relationship between the apparent viscosity and shear rate test data conforms to the power-law equation.

### 3.3. Microstructure Properties of Stigmasterol Organogels

Morphology of the stigmasterol organogels was studied using a polarized light microscope (PLM), XRD, and FTIR.

#### 3.3.1. Polarized Light Microscopy

The three-dimensional network structure is the basis of the mechanical properties of organogels [45]. The influence of gelator concentration on the crystal morphology and microstructure in the organogel system was observed using a polarizing microscope image. The results for the organogels are shown in Figure 7. The stigmasterol crystals were uniformly dispersed in the oil phase, appearing as birefringent patches against a black background [46]. The organogel prepared with 2–4% stigmasterol showed a randomly distributed rod-like crystal structure, while the organogels prepared from 5%–6% stigmasterol showed a rod-like crystal structure with close distribution, the crystal units of stigmasterol formed the three-dimensional structure of the organogels. The number of crystals increased significantly and the internal crystal size of the organogels gradually decreased from random crystal to tightly distributed rod structure with the increase of stigmasterol concentration. However, when the stigmasterol concentration was 7%, the crystal structure of organogels partially overlapped [47]. This is probably because a high oleogelator concentration leads to a higher degree of supersaturation which can accelerate nucleation and restricted the further growth of stigmasterol crystals.

The polarized light results of stigmasterol organogels further confirm that the crystal structure of the organogels can self-assemble to form a compact rod-like fiber structure as the concentration of stigmasterol increases. This structure shows a stronger combination ability of the oil phase and can affect the mechanical resistance of the OBC and create a higher complex module (G’) in the rheological behavior, proving the correlation between the microstructure of the organogel and its mechanical resistance [48]. The crystal network formed by the independent assembly was rearranged when the concentration of stigmasterol reached the supersaturated state, leading to the instability of the organogel structure and the decline of the macroscopic properties. The concentration of the oleogelator plays a key role in controlling the non-covalent interaction-driven self-assembly of fibrillar networks in most cases.

#### 3.3.2. X-ray Diffraction

The microstructure diagram of the gel system can only analyze the changes in crystal units in the gel system. However, the changes in cell parameters and crystal types can be obtained more accurately by XRD analysis. The d-spacing distance in the XRD analysis parameter represents the distance between two diffraction crystal planes of the sample and is used to reflect the crystal type of the sample and the homogeneous polycrystalline phenomenon of fat [49].

The diffraction patterns of rapeseed oil, neat stigmasterol, and organogels prepared with different concentrations of stigmasterol are shown in Figure 8a,b. Two major peaks at 4.52 Å and 4.26 Å were observed in pure stigmasterol and stigmasterol organogels, respectively. In the wide-angle region, the peak around 4.5 Å is usually considered as the characteristic peak of β-polymorph, and peak around 4.2 Å is the characteristic peak of α-polymorph [50]. In other words, the major peaks corresponding to pure stigmasterol and organogels reveal two distinct modes of parallel stacked arrangements, namely α and β. The organogel samples contained the positions and d-spacing of the main peaks corresponding to stigmasterol which indicated that the diffraction pattern of stigmasterol did not change during the formation of organogels. The diffraction patterns of both stigmasterol and organogel samples showed the existence of long and short spacing peaks. According to reports, the presence of long-spacing peaks provides information about the order of the molecular layers, while the presence of short-spacing peaks provides information about the lateral stacking of molecular layers [51]. Compared with the spectra of stigmasterol, the intensity of long-distance peaks in the organogel changed and the positions of some peaks shifted. This indicated that the addition of rapeseed oil in the stigmasterol caused a rearrangement of stigmasterol molecular packaging [52]. In addition, the long-distance peak of organogels enhanced with increasing concentrations of stigmasterol, indicating that the number of crystal structures of organogels increased. This corresponds to the results presented by polarizing microscopes. The variation of the spacing peak of organogels increases with the increasing stigmasterol concentration which further explains the effect of stigmasterol concentration on the structure of organogels.

#### 3.3.3. Fourier Transform Infrared Spectroscopy

FTIR spectroscopy is necessary to understand the interaction between the packing arrangements of organogelator molecules. The FTIR spectra of rapeseed oil, original stigmasterol, and the organogel samples prepared by different concentrations of stigmasterol are shown in Figure 9a,b. The spectra show the absorption bands of organogel owing to the functional groups in rapeseed oil and stigmasterol. The infrared absorption band of rapeseed oil ranged between 400–1800 cm^−1^ and 2800–3100 cm^−1^. The peaks of C-H emerged at approximately 3000 cm^−1^. Furthermore, the bands below 3000 cm^−1^ (2920 cm^−1^) are attributed to the symmetric and anti-symmetric stretching of C-H in -CH_3_ and -CH_2_ functional groups [53], respectively. The characteristic absorption peak around 3346 cm^−1^ is the spectra of original stigmasterol, linked to the stretching of -OH groups [54]. The organogel samples only showed the characteristic peak around 3338 cm^−1^, suggesting that the intermolecular hydrogen bonding observed in the oleogels comes from stigmasterol [55]. Furthermore, new covalent bonds did not form, which is consistent with the results of rheological frequency scanning. However, the characteristic peak of stigmasterol in the organogel samples shifted to a lower wavenumber with the stigmasterol increasing concentration. These results showed that the three-dimensional network structure of organogels was formed by stigmasterol aggregates through intermolecular hydrogen bonding and that the supramolecular aggregates are spontaneously formed through aggregation-nucleation-growth pathways of the stigmasterol crystals [56]. Stigmasterol was a kind of low molecular oleogelator that required the formation of a self-assembled network structure before supramolecular aggregation in an organogel structure could occur. Therefore, the self-assembly pathway of the stigmasterol determined the internal structure of the organogel which further affected its macroscopic properties. This is in line with previous research by Meng, Z et al. which showed that hydrogen bonds may have been responsible for the formation of the crystal structure that fixed the sunflower oil and provided the favorable physical characteristics of the PGE organogel [46]. Similar results were also observed in the SMS-PO organogels [57] reported by Suzuki, M. We thus conclude from the FTIR measurements that the hydrogen bonds play a significant role in the formation of the stigmasterol organogels.

## 4. Conclusions

We prepared the rapeseed oil-based organogels using stigmasterol as a self-assembly oleogelator. The formation of the organogel was related to the gelling temperature and the concentration of the oleogelator which had a critical gelation concentration of 2% at a gelling temperature of 5 °C. The results of macroscopic characteristics showed that the oil holding capacity increased to more than 99.74% when the stigmasterol concentration was 6%. The rheological properties revealed that the organogels prepared with stigmasterol were a pseudoplastic fluid with shear thinning. The results of microscopic characteristic tests showed that the stigmasterol concentration had a significant effect on the integrity and fineness of the crystal structure of the organogels, playing a key role in the densification of crystal network units and crystal size of the organogels. Stigmasterol impregnates rapeseed oil with intermolecular hydrogen bonding to form the crystal network structure, further proving that the concentration of stigmasterol had an important effect on the physical properties and microstructure of the rapeseed oil-based organogels.

This study demonstrates that stigmasterol is a preferable oleogelator to provide an effective approach for the preparation of highly unsaturated organogels. Stigmasterol traps liquid oil in its thermoreversible gel network through gelation, resulting in an organogel with plastic solid properties. Because these liquid oils have specific consistency and hardness without changing their chemical composition, they have great application potential for replacing saturated fats with plastic fats. More extensive applications of rapeseed oil-based organogels should be investigated in order to generate healthier and more spreadable food products in the future.

## Figures and Tables

**Figure 1 foods-11-00939-f001:**
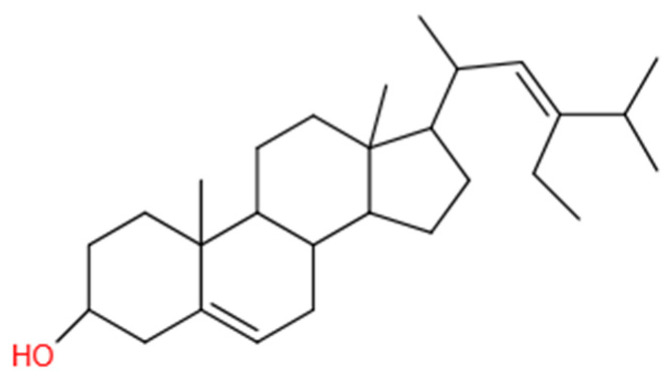
Chemical structure of stigmasterol.

**Figure 2 foods-11-00939-f002:**
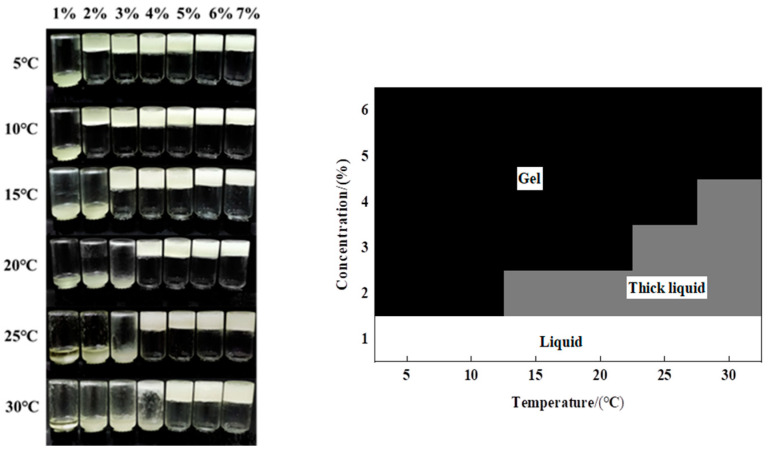
Temperature of Gelation Phase Diagram for different concentrations of stigmasterol organogels at different temperatures: gel (i.e., freestanding gel), thick liquid (liquid was clearly thickened, but freestanding gel was not observed), and liquid (i.e., no gelation observed).

**Figure 3 foods-11-00939-f003:**
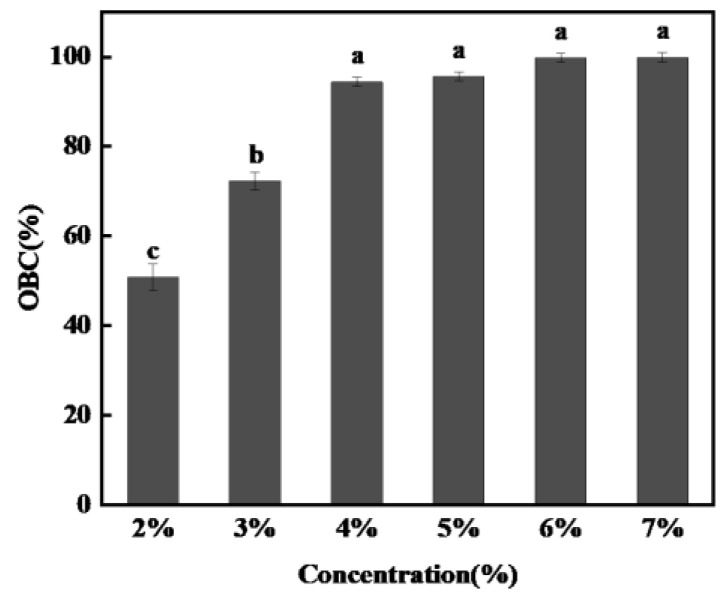
Effect of stigmasterol concentration on the oil binding capacity of the stigmasterol organogels. (Means with different letters in the same classification significantly differ at *p* < 0.05).

**Figure 4 foods-11-00939-f004:**
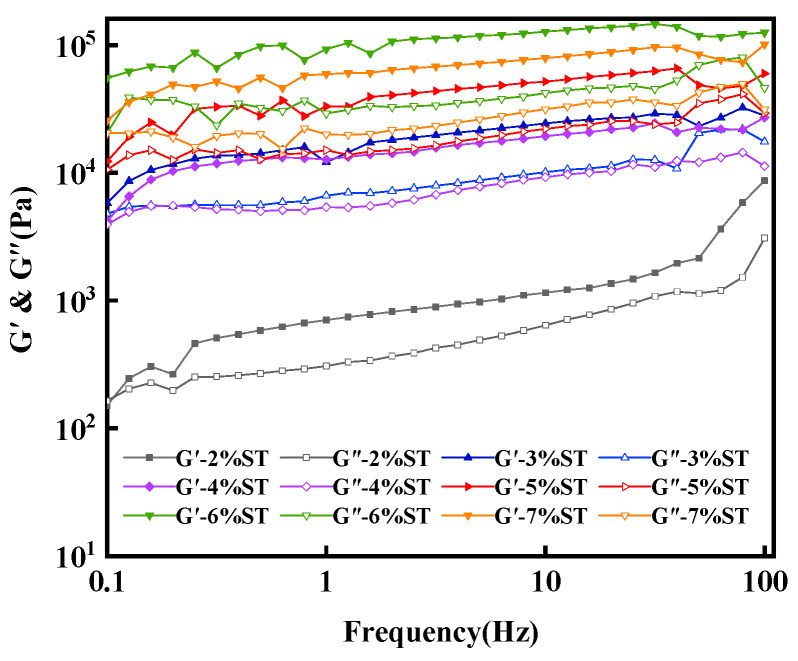
Viscoelastic properties of stigmasterol organogels from the different stigmasterol concentrations (2–7%) with the scanned frequency range from 0.1 Hz to 100 Hz at 25 °C.

**Figure 5 foods-11-00939-f005:**
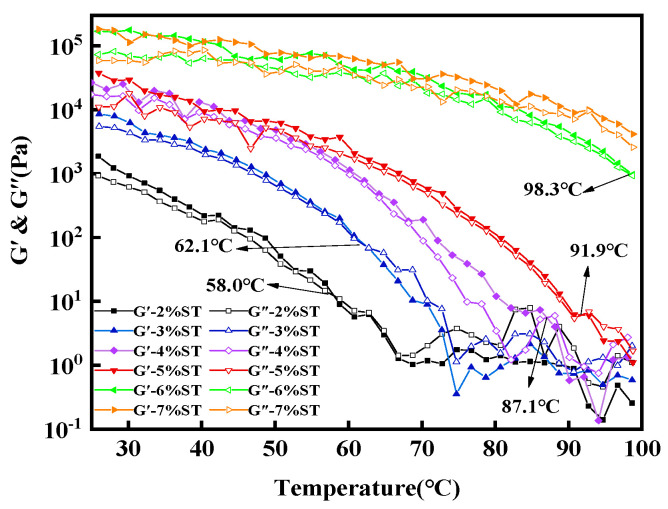
Different stigmasterol concentrations (2–7%) of stigmasterol organogels had viscoelastic properties in the temperature range of 25 to 100 °C.

**Figure 6 foods-11-00939-f006:**
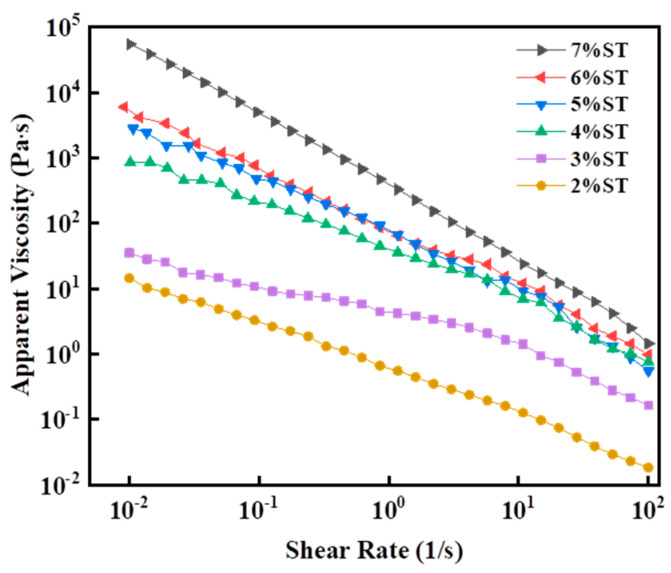
Changes in the apparent viscosity of stigmasterol organogels with different stigmasterol concentrations (2–7%) and shear rates ranging from 0.01 s^−1^ to 100 s^−1^.

**Figure 7 foods-11-00939-f007:**
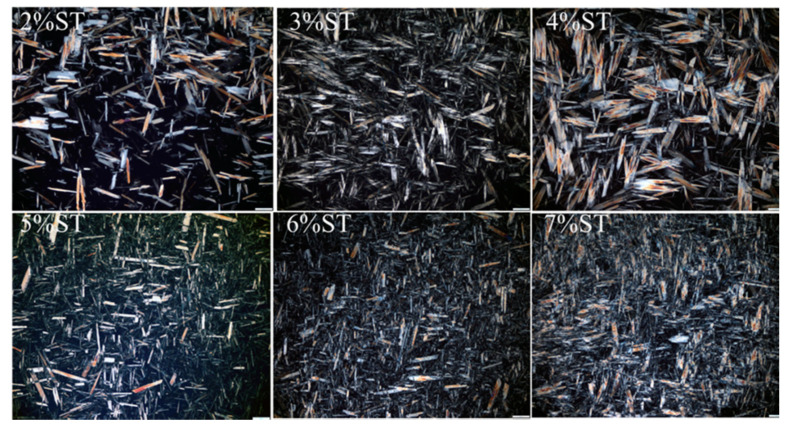
Microstructures of stigmasterol organogels observed at 100× after 24 h storage at 5 °C.

**Figure 8 foods-11-00939-f008:**
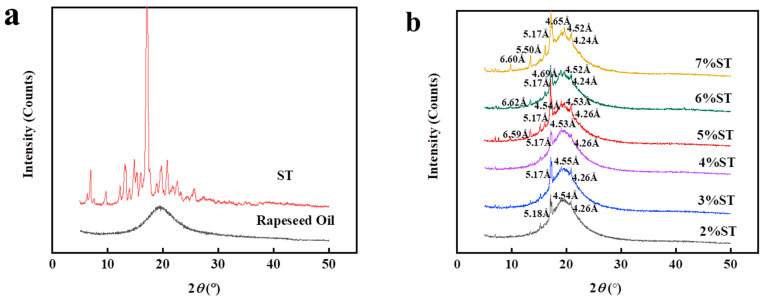
The XRD patterns of rapeseed oil and stigmasterol (**a**); organogels with diverse stigmasterol concentrations (**b**).

**Figure 9 foods-11-00939-f009:**
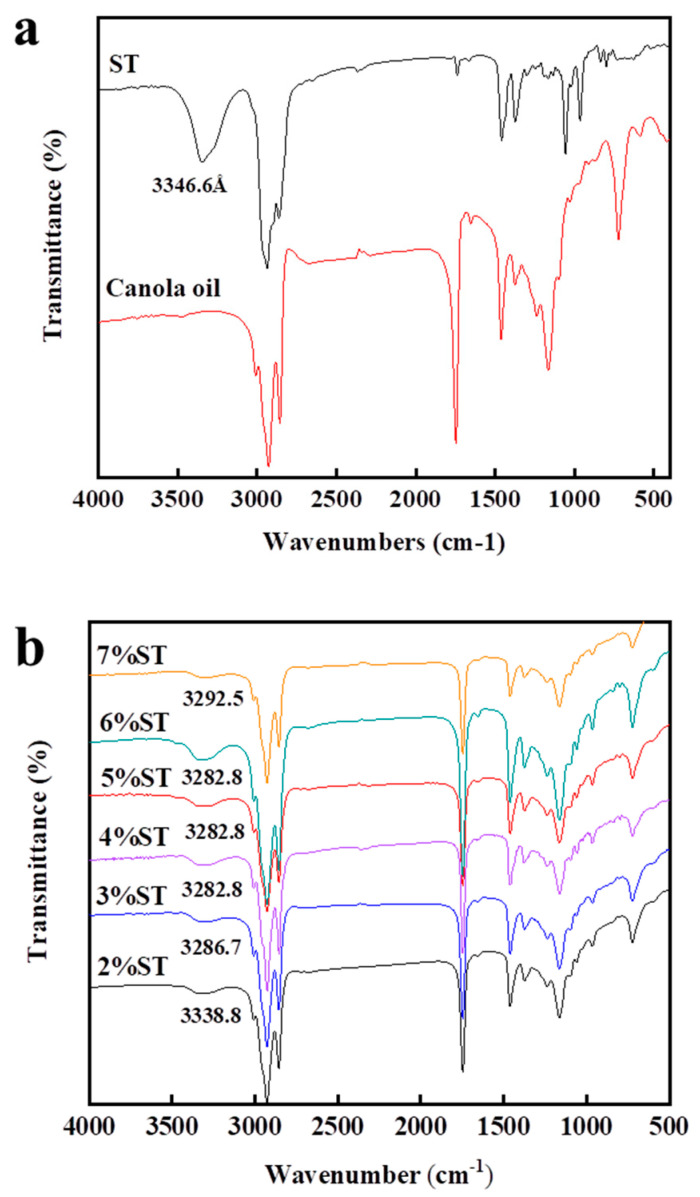
The FT-IR of rapeseed oil and stigmasterol (**a**); organogels with diverse stigmasterol concentrations (**b**).

**Table 1 foods-11-00939-t001:** Values of power-law parameters (K, n) of the stigmasterol organogels at different stigmasterol concentrations in the range of 0.01–100 s^−1^.

ST Concentration	K/Pa·s	n	R^2^
2%	0.42 ± 0.027	0.47 ± 0.087	0.997
3%	12.71 ± 0.088	0.42 ± 0.068	0.991
4%	39.49 ± 0.069	0.25 ± 0.025	0.991
5%	63.66 ± 0.032	0.14 ± 0.017	0.998
6%	81.59 ± 0.019	0.08 ± 0.022	0.997
7%	253.60 ± 0.048	0.03 ± 0.046	0.999

Note: Values are means ± standard of deviations.

## Data Availability

Data is contained within the article.

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
