# Peer review of "Structure and Properties of Organogels Prepared from Rapeseed Oil with Stigmasterol"

_foods, 2022, doi:10.3390/foods11070939_

Round 1

Reviewer 1 Report

This is an interesting manuscript in which the authors characterize the properties of organogels prepared by mixing rapeseed oil with stigmasterol.  Overall, the manuscript is well written (with occasional typographical errors), but the novelty of the manuscript is not apparent.  I have several comments/suggestions I would like to ask the authors to address below.

--The authors provide only the chemical name but no chemical structure of stigmasterol.  It is unclear how the stigmasterol binds with unsaturated fatty acids.  For example, what functional group of stigmasterol binds with the fatty acids moieties, and what intermolecular interactions could promote this binding?

--L 182.  Please explain what intermolecular interaction(s) would force the crystal formation.  Why does stigmasterol's OBC value plateau at 7%(w/w)?

--The experimental section should indicate how the G’ and G” were calculated within the defined frequency range.

--L219-221.  This is a poorly worded sentence.  Please reword with clarity.

--L292.  I do not understand what is meant by “those denser crystal distributions-----structural network…..” How this would ultimately benefit the properties of organogels prepared this way.

--XRD pattern shows two peaks at 4.5 0A and 4.2 0A, which are assigned to stigmasterol and stigmasterol organogels, respectively.  Please clarify the origin of peaks at 5.170A, 5.5 0A, and 6.5 0A.  Why are more peaks seen upon increasing the concentration of stigmasterol in the XRD pattern?

--Not all bands in FTIR are properly assigned

--L368.  Note the part of the sentence “-----hydrogen bonds may were responsibility for the formation of the crystal-------“ need correction with the proper writing style

--If hydrogen bond plays a key role the draw the chemical structure with hydrogen bond acceptor and donor sites on stigma sterols and fatty acids.  Finally, are animal or human studies performed showing stigmasterol is more beneficial when replacing saturated fatty acids from rapeseed oil?.  

Author Response

Thank you very much for your suggestions. I have answered some of your questions in the attachment. Please see the attachment  

Reviewer 2 Report

Overall the paper has novelty findings, scientific and well presented.

Only minor to final grammar check by the publisher and strictly reduce the similarity relatively high 30%.

The purpose of the research to understand the physical and microscopic properties of different stigmasterol concentrations on the oil binding capacity. This is relevant for future to better understanding organogels properties.

Not Many research to discussed and research the  physical and microscopic properties of different stigmasterol concentrations. So this is interesting to others researcher and industrial to develop new organogels with different concentration to optimized the physical and microscopic properties.

The manuscript well written and easy to read but my point is the quite high similarity index by Turnitin (attached on system). Please the authors should be revised it.

The  results presented well by the data in figures and tables, but and also supported by relevant update publication for the background and the results obtained.  

Author Response

     Thank you very much for your advice.  The duplication report you provided is very important to me.  I have reduced the weight according tohe duplicate check report you provided, and modified some of its grammar and sentences.  I have submitted the manuscript in the form of attachment, please check the attachment.  

Reviewer 3 Report

Tang et al. described studies on the structure and properties of organogels prepared from rapeseed oil and stigmasterol.

Minor revisions:

Paragraphs 3.1., 3.2.1., 3.2.2., 3.3.1., 3.3.2., 3.3.3.: The texts mentioning the figures should proceed the figures.

Figure 3. Add the temperature value at which the frequency was scanned.

Author Response

Thanks for your advice,  we checked the relevant articles of ''Foods Journal'' and found that their drawings were not marked according to the legend (such as 3.1., 3.2.1., 3.2.2., 3.3.1., 3.3.2., 3.3.3).  So we would like to confirm with you whether it needs to be modified. At the same time, we have modified line 210 of the manuscript in response to your suggestion to add the temperature value when frequency scanning in Figure 3 . 
We have uploaded the manuscript in the form of attachment, please check the attachment